# Single Angio-Seal™ Device as a Simplified and Technically Feasible Alternative for Tract Closure in Percutaneous Portal Vein Access: A Retrospective Study

**DOI:** 10.3390/diagnostics15101266

**Published:** 2025-05-16

**Authors:** Ismail Karluka, Mustafa Mazıcan

**Affiliations:** Department of Interventional Radiology, Adana Dr. Turgut Noyan Application and Research Center, Baskent University, Adana 01250, Turkey; m_mazican@yahoo.com

**Keywords:** Angio-Seal™, portal vein, vascular closure device, percutaneous transhepatic access, hemostasis, portal vein thrombosis, technical success, interventional radiology, tract embolization, procedure-related complications

## Abstract

**Purpose:** This study aimed to evaluate the efficacy and safety of the Angio-Seal™ VIP vascular closure device (VCD) in achieving hemostasis following percutaneous transhepatic portal venous interventions. **Methods:** This retrospective study evaluated 20 patients (mean age: 52.85 ± 16.18 years; 80% male) who underwent percutaneous transhepatic portal vein interventions followed by tract closure with the Angio-Seal™ device between January 2016 and September 2024. Procedural data, pre- and post-procedural hemoglobin and hematocrit levels, and complications were analyzed. Technical success was defined as the successful deployment of the device with immediate hemostasis and no evidence of bleeding on post-procedural imaging. **Results:** Technical success, as defined in this study, was achieved in all 20 procedures (100%). The mean hemoglobin level declined from 11.91 ± 2.01 g/dL to 11.09 ± 2.19 g/dL (*p* < 0.001), and the mean hematocrit level decreased from 36.18 ± 6.03% to 32.98 ± 5.80% (*p* = 0.001). A hemoglobin drop ≥2 g/dL occurred in two patients (10%) and a hematocrit drop ≥4% in six patients (30%); none were associated with imaging or clinical evidence of hemorrhage. No major complications were observed. Minor complications, including localized pain managed with analgesics, occurred in five patients (25%). Follow-up imaging confirmed the absence of hemoperitoneum or device-related failure. **Conclusions:** Angio-Seal™ is a technically feasible, safe, and effective option for tract closure following percutaneous transhepatic portal vein access. This single-device approach may offer a cost-effective alternative to traditional embolization techniques. However, more extensive prospective studies are required to validate these findings.

## 1. Introduction

Many procedures require percutaneous transhepatic portal venous intervention, including partial portal venous embolization before hemihepatectomy, pancreatic islet cell transplantation, transhepatic varicose vein treatment, and transjugular intrahepatic portosystemic shunt placement occasionally [1,2]. Approximately 30% of patients experience bleeding after the procedure is terminated without closing the tract [2]. In some cases, this bleeding may be life-threatening and result in severe morbidity and mortality [1].

Several embolic materials have been used in the literature to prevent bleeding from the tract, including sponge particles, N-butyl cyanoacrylate, coils, and plugs [1,2,3]. These agents have significant disadvantages, such as involuntary portal vein embolization during embolization and insufficient filling of the tract [2,4,5,6].

Angio-Seal™ VIP (Terumo Interventional Systems, Somerset, NJ, USA) is a well-known collagen-based vascular closure device (VCD), which was approved in 1996 for the first time. It consists of a T-shaped bioabsorbable polymer anchor with diameters of 6F and 8F. The device creates a mechanical seal by compressing the artery insertion site between a bioabsorbable anchor and a soluble collagen sponge [7]. Clinical experiences have revealed that Angio-Seal™ VIP is highly effective and safe when closing transfemoral arterial punctures [7].

With the increase in minimally invasive procedures, different techniques have been developed for the safe and effective closure of percutaneous vascular accesses. One of these methods, the Angio-Seal™ vascular closure system, is notable for its rapid and reliable hemostasis without the need for fluoroscopy. This system, which is widely used in the arterial system, stands out due to its low risk of complications. A limited number of studies in the literature have reported its use for portal vein access hemostasis [8]. It has also been reported to have been successfully applied “off-label” in different clinical situations [9,10].

However, except for a limited number of case reports, none of the studies has evaluated the effectiveness of Angio-Seal™ for percutaneous portal venous interventions. For this reason, the primary objective of this study was to evaluate the efficacy and safety of the Angio-Seal™ device as a VCD in percutaneous transhepatic portal venous interventions. This study evaluated whether Angio-Seal™ can be a safe, technically feasible, and clinically effective alternative for tract closure in percutaneous portal venous interventions. Given the lack of standardized closure techniques and the limited literature on dedicated VCDs in this context, Angio-Seal™ may represent a promising option with the potential to simplify the procedure, minimize the risk of bleeding, and improve overall procedural safety.

## 2. Materials and Methods

### 2.1. Patient Population

This study on human participants adhered to the ethical standards established by the institutional research committee and the 1964 Helsinki Declaration, along with its subsequent amendments. This study was approved by the Institutional Review Board of Baskent University (Project no: KA21/218) and supported by the Baskent University Research Fund. This retrospective study involved 20 patients (16 males and 4 females; mean age of 56 [52.85 ± 16.18]) years, who received the Angio-Seal™ device for hemostasis after portal vein interventions between January 2016 and September 2024. The study data were retrieved from the hospital’s electronic medical record system and our department’s database, and included clinical, laboratory, and imaging findings.

The most common indication for portal vein interventions in this study population was acute portal vein thrombosis (PVT) (65%, *n* = 13). Other indications included preoperative portal vein embolization (PVE) to induce hypertrophy of the future liver remnant prior to right hemihepatectomy (20%, *n* = 4). Portal vein stenosis (5%, *n* = 1) was also reported.

Less frequent indications comprised combined stenosis of the main portal vein and splenic vein (5%, *n* = 1), while in one patient (5%, *n* = 1), left gastric vein embolization for recurrent esophageal variceal bleeding was performed after the failure of conventional therapies. A wide range of endovascular procedures, tailored to the vascular pathology and clinical indication in each patient, were performed in our study (Table 1).

In all cases, the indication for portal venous intervention was determined through a multidisciplinary consensus among interventional radiologists, gastroenterologists, and general surgeons following the evaluation of the patient’s clinical status and imaging findings.

The most commonly employed technique was manual thromboaspiration combined with catheter-directed thrombolytic therapy, which was performed in seven patients (35%). Manual thromboaspiration with percutaneous transluminal angioplasty (PTA) was applied in three patients (15%), and PTA alone was performed in another three patients (15%). Embolization procedures were performed in five patients (25%), while thrombolytic therapy alone was administered to two patients (10%).

Percutaneous access was performed through the right side in 80.0% of the patients, while the left side was used in 20.0%. The mean INR value before the procedure was 1.33, with a range from 1.05 to 2.54. The mean prothrombin time (PT) was 23.1 s, with a minimum of 12.2 s and a maximum of 90.2 s. The mean pre-procedural platelet count was 270.2 × 10^3^/μL, with a range from 75.0 × 10^3^/μL to 681.0 × 10^3^/μL.

Although most patients had INR and platelet values within or near normal limits, in cases with an elevated INR or thrombocytopenia, coagulation parameters were corrected with supportive treatment, such as fresh frozen plasma, vitamin K, or platelet transfusion prior to the procedure.

It is important to note that the Angio-Seal™ VIP device contains a bioabsorbable collagen sponge derived from bovine sources (Achilles tendon). Therefore, its use is contraindicated in patients with known hypersensitivity to bovine-derived materials or collagen. All patients included in this study were screened for any history of collagen allergy prior to device deployment.

### 2.2. Technique for Implementing the Angio-Seal™ Device in Portal Vein Access

In this technique, the procedure was carried out while the patient was under intravenous sedation. All procedures were performed under intravenous sedation administered by an anesthesiologist. The sedation regimen, selected and titrated at the anesthesiologist’s discretion, typically included midazolam (0.02–0.05 mg/kg) and fentanyl (1–2 mcg/kg). In some cases, propofol (0.5–1 mg/kg bolus) was used as needed for deeper sedation. Local anesthesia was applied to the incision area. The Accustick™ II introducer system (Boston Scientific Corporation, Marlborough, MA, USA) was utilized to access the peripheral right portal vein branch. This procedure was conducted under ultrasonography (USG) and fluoroscopy for guidance. The vascular sheath was inserted into the portal vein. The diameter of the vessel sheath, chosen according to the procedure to be performed and the preference of the administering operator, ranged from a minimum of 5F to a maximum of 8F. Subsequently, the treatment was commenced via the portal vein. Following the completion of the procedure, access to the portal vein was sealed to maintain hemostasis.

To accomplish this, 0.035 stiff guidewires (Amplatz Super Stiff Straight Medicine Guidewire 260 cm, Boston Scientific Cooperation, Heredia, Costa Rica) were inserted through the vascular sheath. The vessel sheath was meticulously and wholly extracted. The guidewire was carefully observed under fluoroscopy to ensure its safe positioning inside. As part of the procedure to close the vessel, the Angio-Seal™ device was used. The Angio-Seal™ standard technique was employed. Each step was observed under fluoroscopy. While the guidewire was held in place, the vascular sheath was removed. The first puncture was replaced with an Angio-Seal™ localizer system suitable for the vascular sheath’s diameter (6F or 8F). Blood flow was observed through the locator and the proper sheath position was visually verified. The Angio-Seal™ device was inserted into the sheath until it “clicked”. The locking cap was slowly pulled back until another “click” sounded. The device anchor was locked in place. The Angio-Seal™ VIP device was then slowly retracted until the suture coil stopped. Upward tension was maintained on the device and the compression tube was slowly advanced until resistance was felt. Then, the seam was cut and the device removed (Figure 1 and Figure 2). No other embolic agents were used to block the pathway. After these stages, the tract level was checked with the USG. The presence of hemoperitoneum was then evaluated.

### 2.3. Follow-Up

Following the procedure, all patients underwent an initial bedside USG evaluation to exclude hemoperitoneum. Once the absence of hemoperitoneum was confirmed, a second routine USG examination was performed at the 24th hour in all patients. All ultrasonographic evaluations were conducted by radiologists with at least 10 years of experience in abdominal ultrasonography, ensuring consistency and reliability in the imaging assessment.

If no hemoperitoneum was detected on the USG, but there was a hemoglobin decrease greater than 2 g/dL or a hematocrit drop exceeding 4%, contrast-enhanced computed tomography (CT) was performed for further evaluation. This imaging aimed to determine whether the observed drop was related to intraprocedural blood loss or hemodilution, or if it was indicative of a post-procedural complication. Additionally, CT imaging was utilized to assess the treatment’s effectiveness and exclude other potential complications.

In addition to laboratory values, patients were closely monitored for clinical signs, such as abdominal pain, hypotension, tachycardia, and peritoneal irritation. The follow-up period for each patient was determined based on either the date of death or their most recent hospital admission, as recorded in the clinical documentation.

Technical success was defined as the successful deployment of the Angio-Seal™ VIP device with the confirmed placement of the localizer and anchor within the portal vein, appropriate suture tension indicating anchor engagement, and completion of the procedure with the advancement of the collagen plug into the liver parenchyma. Additionally, the absence of hemoperitoneum on immediate post-procedural USG was required to confirm technical success.

Complications were evaluated based on the patient’s medical records and post-procedure imaging—classification of complications adapted from the Society of Interventional Radiology (SIR) guidelines [11]. Major procedural complications were hemoperitoneum, biliary peritonitis, migration of collagen plaque to the portal vein, hemobilia, hepatic artery injury, severe liver damage, and permanent sequelae death. Mild pain that could be controlled with medical therapy was defined as a minor complication.

### 2.4. Statistical Method

Descriptive statistics were used to summarize the clinical and laboratory data. Paired sample t-tests were applied to compare pre-and post-procedural hemoglobin and hematocrit levels. Numerical variables were expressed as mean ± standard deviation (SD) and median (range), while categorical data were presented as numbers and percentages. A *p*-value < 0.05 was considered statistically significant.

## 3. Results

A total of 20 patients underwent percutaneous transhepatic portal venous interventions followed by tract closure using the Angio-Seal™ device. Of these, 16 (80%) were male and 4 (20%) were female. The mean age was 52.85 ± 16.18 years (range: 15–79 years), with a median age of 56 years.

### 3.1. Procedural Details and Technical Success

Technical success, according to the procedural definition described in Section 2 (Materials and Methods), was achieved in all 20 procedures (100%). Depending on the clinical indications, thromboaspiration was performed in 12 patients (60%), percutaneous transluminal angioplasty (PTA) in 6 patients (30%), thrombolytic therapy in 8 patients (40%), and embolization in 5 patients (25%). As some patients underwent more than one procedure, total percentages exceeded 100%.

### 3.2. Laboratory Outcomes

The mean hemoglobin level decreased from 11.91 ± 2.01 g/dL before the procedure to 11.09 ± 2.19 g/dL afterward, with a mean difference of 0.82 ± 0.86 g/dL (*p* < 0.001). Similarly, the mean hematocrit dropped from 36.18 ± 6.03% to 32.98 ± 5.80%, with a mean difference of 3.2 ± 3.49% (*p* = 0.001). A hemoglobin decrease ≥ 2g/dL was observed in two patients (10%), while a hematocrit decrease ≥ 4% occurred in six patients (30%). (Table 2).

### 3.3. Complications and Follow-Up

No major complications were observed. Minor complications were reported in five patients (25%) and were defined as right upper quadrant or access site-related pain persisting beyond 24 h after the procedure. All cases responded well to analgesic treatment and required no further intervention.

In all six patients with a hematocrit drop ≥4%, contrast-enhanced CT was performed to rule out post-procedural complications. None of the CT scans revealed active bleeding, hemoperitoneum, or Angio-Seal™-related device failure.

No patient developed hypotension, peritoneal irritation, or other clinical signs requiring additional imaging or intervention. The mean follow-up duration was 206.3 ± 257.9 days, with a median of 95.5 days (19–805 days) (Table 3).

These findings suggest that the Angio-Seal™ VIP device provides adequate and durable hemostasis without increasing the risk of post-procedural complications during follow-up.

Although follow-up periods varied widely, with some patients followed beyond two years, no delayed access site-related complications were encountered.

## 4. Discussion

The findings of the present study highlight our initial experiences with a limited number of patients, demonstrating that Angio-Seal™ is a viable and effective method for sealing puncture sites in portal vein procedures. We observed no significant bleeding from the portal access points sealed with Angio-Seal™, and no major complications were associated with its use. This research provides valuable insights from preliminary trials with a small patient cohort using Angio-Seal™ for achieving hemostasis after portal vein interventions.

Ultrasound and fluoroscopy typically guide transhepatic portal vein interventions [2]. While this method is generally safe, it can lead to minor issues like mild hypotension from pain or vasovagal reactions at the puncture site. More severe complications may include bleeding, hemobilia, pseudoaneurysm, or infection. Bleeding, often venous and stemming from the portal vein, can be serious and potentially life-threatening, causing severe hemodynamic instability [12]. Various materials, including gelatin sponge particles, biological adhesives, coils, and vascular plugs, have been employed to seal the puncture site after interventions [12,13,14]. Coils and gelatin sponge particles are the most frequently used but have drawbacks. Gelatin sponge particles might not completely close the duct or can lead to delayed bleeding as they dissolve over time.

On the other hand, coils are more complex to use, may require multiple applications for complete embolization, and carry a risk of incomplete closure. Incorrectly sized coils can also be displaced into the portal vein or abdominal cavity—other materials, such as NBCA, present risks, including incomplete or unintended embolization [2,15]. Vascular plugs, while effective, involve a lengthy procedure, potential for distal displacement similar to coils [1,16], and tend to be more expensive than other options [1]. Furthermore, there has not been any evidence that these agents are superior in efficacy and safety. All the agents used have advantages and disadvantages [6].

A recent comparative study by Marra et al. [6] reported 18 post-procedural bleeding events among 220 cases, with no statistically significant differences between patients treated with coils, NBCA, or gelfoam. However, bleeding events were more frequently observed in patients treated with coils or who did not undergo tract embolization, suggesting that while all techniques are generally effective, bleeding risk is not entirely eliminated. In contrast, our study demonstrated no clinically or radiologically evident bleeding following single-device Angio-Seal™ closure, suggesting comparable or potentially superior hemostatic control when applied to selected portal vein access tracts.

Advancements in medical technology have led to the development of various active VCDs, each with unique structures and applications. Among these are VCDs, such as Cardiva Catalyst (Cardiva Medical, Inc., Sunnyvale, CA, USA), which have garnered attention for their innovative designs. In collagen plug-closure devices, products such as the Angio-Seal™ VIP (Terumo Interventional Systems, Somerset, NJ, USA) and the Mynx (AccessClosure, Mountain View, CA, USA) stand out. The category of VCDs utilizing polyglycolic acid plugs includes notable examples, like ExoSeal (Cordis Corporation, Miami Lakes, FL, USA). Additionally, there are field-features clipping devices, such as Starclose (Abbott Vascular, Redwood City, CA, USA), and suturing devices, as exemplified by Perclose (Abbott Vascular), highlighting the diverse range of closure mechanisms available in modern medical practice [17]. The literature has documented the use of VCDs in portal venous interventions. In their study, Tan et al. [2] explored the effectiveness of the Mynxgrip^®^ VCD for such procedures. Their technique involved initially sealing the pathway between the portal vein and the surface of the liver parenchyma using the Mynxgrip^®^ VCD. Subsequently, they employed an embolization process, using either n-butyl cyanoacrylate (NBCA) or a dense gelatin paste, to close off any remaining channels within the parenchyma. This approach has been demonstrated to effectively and safely seal percutaneous transhepatic portal venous access tracts, even in patients with bleeding diathesis.

A study by Adani et al. [18] reported one of the first instances of using a collagen-based VCD to seal a portal vein. In this study, early PVT developed in three patients following major liver surgery. The focus was on demonstrating the effectiveness of minimally invasive percutaneous transhepatic portography for treating PVT. The procedure involved the mechanical fragmentation and pharmacological lysis of the thrombus. The closure of the percutaneous portal vein access point is a critical aspect. This closure was achieved using two collagen cylinders (Vaso Seal Vascular Hemostasis Device; Datascope Corp., Montvale, NJ, USA).

Pescatori et al. [19] showcased the application of the Angio-Seal™ device in two cases of transhepatic portal vein stent-graft implantations. The procedures were uniquely concluded by deploying two 8 Fr Angio-Seal™ closure devices. This involved a coordinated approach where one operator tensioned the suture while another inserted the collagen plug into the liver parenchyma. Their report highlights the successful and safe use of percutaneous closure devices in managing transhepatic portal access.

An important technical consideration during Angio-Seal™ deployment in transhepatic procedures is the length of the sheath relative to the hepatic parenchymal tract. While the 11 cm sheath was sufficient in our adult cohort, it may exceed the available tract in pediatric patients or individuals with reduced liver volume. Inadequate parenchymal depth can compromise anchor engagement and increase the risk of peritoneal deployment or bleeding. Therefore, a careful pre-procedural imaging assessment is essential for appropriate patient selection and device safety, as highlighted in recent case series and anatomical analyses [8].

An essential technical aspect in our approach was the deliberate selection of peripheral portal vein branches for vascular access. Anatomical considerations guided this choice, as peripheral branches provide superior parenchymal support for anchor engagement compared to larger, straighter central segments. The Angio-Seal™ device used in our study incorporates a T-shaped bioabsorbable anchor, which in the 6F version measures approximately 10 mm in length and 1.2 mm in width [20]. While the manufacturer does not disclose the exact dimensions of the anchor in the 8F version, it is presumed to be proportionally larger to match the increased sheath diameter. Pre-procedural cross-sectional imaging was crucial to ensure that the target vessel had a compatible lumen size and that the hepatic parenchyma could provide sufficient resistance for secure anchor deployment. This anatomical and technical alignment likely contributed to our series’s 100% technical success rate.

Although a formal cost analysis was beyond the scope of this study, several practical inferences can be drawn. Using a single Angio-Seal™ device simplifies the tract closure process by eliminating the need for multiple embolic agents or adjunctive maneuvers, which are often required with traditional techniques, such as coils or NBCA. In the comparative study by Zhang et al. [15], the average number of coils used per case was 1.2 ± 0.4. NBCA was used in a mean volume of 0.35 ± 0.07 mL, with both techniques necessitating continuous fluoroscopic monitoring and precise catheter manipulation [15]. These methods may increase procedural time and operator workload, indirectly contributing to higher overall costs, especially in long or tortuous tracts that require more materials or repositioning. Although the unit cost of the Angio-Seal™ device may be higher than individual embolic agents, the total procedural cost, when considering material use, procedure duration, and post-procedural care, may be comparable or even favorable in selected cases. Therefore, while our initial findings suggest potential cost-effectiveness, prospective economic analyses across diverse clinical settings are warranted.

Angio-Seal™ was used as the vascular closure device in the patients in the present study. Our initial experience shows that the collagen plug between the portal vein and the liver parenchyma surface provides good hemostasis. This technique is similar to transfemoral closure in portal vein interventions via the transhepatic route. Clinical, laboratory, and follow-up images were not compatible with portal vein access tract bleeding in any of the patients. Technical success was found to be 100%. The intensive use of Angio-Seal™ in transfemoral artery interventions and the fact that the peripheral portal vein branch can be accessed under the guidance of USG and fluoroscopy contributed to this high degree of technical success. It also reduced the risk of complications.

The technique used in the present study was different from others discussed in the literature. While the double Angio-Seal™ technique, which typically uses an 8F and often requires two operators, is common, it is also costly. Another method involves using a single collagen plug with a VCD, followed by another embolizing agent, but this approach is technically complex and can increase costs. Our research indicates that it is feasible to use a single Angio-Seal™, and that this is also safe. This technique is familiar to most interventional radiologists, who commonly use the Angio-Seal™ VIP or similar devices to close the femoral entry site. We have shown that it is also effective for portal vein interventions, as demonstrated here with a small patient group. While traditional tract closure methods have advantages, this study suggests that using a single Angio-Seal™ is a cost-effective alternative, especially for interventional radiologists experienced in transhepatic portal interventions. This method could offer a valuable alternative to traditional and sometimes disadvantageous techniques.

## 5. Limitation

The most important limitations of the present study are the small number of patients and the retrospective design. This limitation is significant because it restricts the ability to generalize the findings and conclusively demonstrate the superiority of the Angio-Seal™ device over other methods. While the initial results are promising, showing high technical success and safety, the small sample size means these findings are preliminary. To definitively establish the advantages of Angio-Seal™, larger-scale studies with more participants must provide robust, statistically significant data that can validate its efficacy and safety compared to other closure techniques in portal vein interventions. In addition, systemic markers of coagulation activation (e.g., D-dimer, thrombin-generation assays, Prothrombin Fragment 1 + 2) were not systematically measured. Because most patients presented with acute portal vein thrombosis and most patients received immediate anticoagulation, these biomarkers would not have allowed the attribution of sub-clinical thrombotic activation specifically to the closure device.

## 6. Conclusions

In conclusion, this preliminary study with a small patient cohort suggests that the Angio-Seal™ is a promising and effective method for achieving hemostasis in percutaneous portal venous interventions. The technique demonstrated high technical success and safety, with no major complications or hemoperitoneum post-procedure. However, due to the study’s small size and retrospective design, these findings are preliminary, and further research with a larger patient population is needed to confirm the efficacy and safety of the Angio-Seal™ device compared to other closure techniques in portal vein interventions.

## Figures and Tables

**Figure 1 diagnostics-15-01266-f001:**
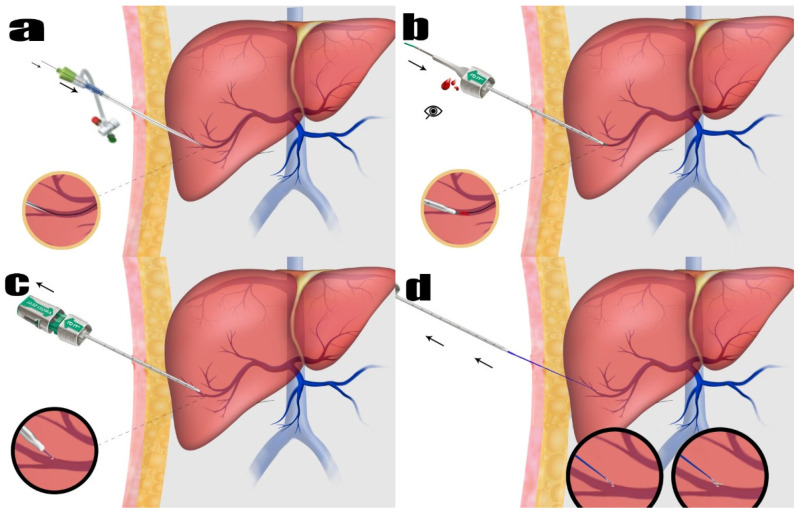
Portal vein access and Angio-Seal™ procedure. (**a**) The procedure begins with the catheterization of the portal vein using a vascular sheath and guidewire, establishing transhepatic portal vein access. (**b**) Following the removal of the vascular sheath, the Angio-Seal™ localizer system is positioned over the guidewire. Blood flow within the localizer system confirms the correct placement in the portal vein. (**c**) The Angio-Seal™ device is inserted into the sheath until a “click” indicates proper placement. The locking cap is gradually retracted until another “click” is heard, signifying that the T-shaped anchor has been deployed within the portal vein lumen. (**d**) Angio-Seal™ is carefully withdrawn until the suture coil halts. The compression tube is gently pushed forward while maintaining upward tension on the device until resistance is encountered. The suture is then cut, and the device is entirely removed, completing the procedure.

**Figure 2 diagnostics-15-01266-f002:**
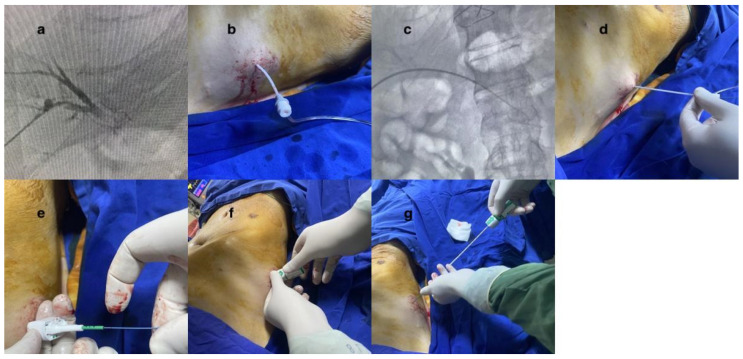
Stepwise illustration of the Angio-Seal™ technique. (**a**) Accessing a peripheral portal vein branch: Utilizing the Accustick™ II introducer systems to gain access to a branch of the peripheral portal vein. (**b**) Vascular sheath introduction: Placement of a vascular sheath into the vein. (**c**) Fluoroscopic guidewire insertion: Observing the vascular sheath and inserting a guidewire through it, using fluoroscopy for visual guidance. (**d**) Vascular sheath removal: Carefully removing the vascular sheath from the vein. (**e**) Angio-Seal™ localizer and blood flow demonstration: Showcasing the functionality of the Angio-Seal™ localizer and monitoring blood flow. (**f**) Angio-Seal™ placement and locking: Positioning the Angio-Seal™ device and securing it in place until a “click” sound is heard, indicating proper locking. (**g**) Suture stretching and collagen plug placement: Pulling the entire system to stretch the suture and subsequently placing the collagen plug for secure closure.

**Table 1 diagnostics-15-01266-t001:** Demographic and clinical characteristics.

Parameter	Details
Total Patients	20
Gender	16 Male (80%), 4 Female (20%)
Mean Age	52.85 ± 16.18 years
Access Side	Right (80%), Left (20%)
Most Common Indication	Acute PVT (65%, *n* = 13)
Other Indications	Preoperative PVE (20%, *n* = 4), Portal vein stenosis (5%), Combined stenosis (5%), Left gastric vein embolization (5%)

**Table 2 diagnostics-15-01266-t002:** Procedural and laboratory details.

Parameter	Details
Techniques Used	Thromboaspiration + Thrombolysis (35%, *n* = 7), Thromboaspiration + PTA (15%, *n* = 3), PTA alone (15%, *n* = 3), Embolization (25%, *n* = 5), Thrombolysis alone (10%, *n* = 2)
Mean INR	1.33 (range: 1.05–2.54)
Mean PT (s)	23.1 (range: 12.2–90.2)
Mean Platelet Count (×10^3^/μL)	270.2 (range: 75.0–681.0)
Pre-Procedure Hemoglobin (g/dL)	11.91 ± 2.01
Post-Procedure Hemoglobin (g/dL)	11.09 ± 2.19
Hemoglobin Decrease ≥ 2 g/dL	2 Patients (10%)
Pre-Procedure Hematocrit (%)	36.18 ± 6.03
Post-Procedure Hematocrit (%)	32.98 ± 5.80
Hematocrit Decrease ≥4%	6 Patients (30%)

**Table 3 diagnostics-15-01266-t003:** Outcomes, complications, and follow-up.

Parameter	Details
Technical Success	100%
Major Complications	None observed
Minor Complications	5 patients (25%)—pain controlled with analgesics
CT Findings (if HCT↓ ≥4%)	No bleeding or device failure detected
Hemodynamic Instability	None observed
Follow-Up Duration	206.3 ± 257.9 days (median: 95.5 days)
Conclusion	Angio-Seal™ device was safe and effective; further studies are needed

## Data Availability

The data supporting the reported results of this study are not publicly available due to privacy and ethical restrictions. However, de-identified data may be made available from the corresponding author upon reasonable request and with permission from the Institutional Review Board of Başkent University.

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
