# Peer review of "Single Angio-Seal™ Device as a Simplified and Technically Feasible Alternative for Tract Closure in Percutaneous Portal Vein Access: A Retrospective Study"

_diagnostics, 2025, doi:10.3390/diagnostics15101266_

Round 1

Reviewer 1 Report

Comments and Suggestions for Authors

The authors have done a good job in writing this manuscript on an important topic entitled "Single Angio-Seal™ device as a simplified and technically feasible alternative for tract closure in percutaneous portal vein access: A retrospective study".  The authors did this retrospective study in 20 patients.  The patients underwent percutaneous transhepatic portal vein interventions followed by tract closure with Angio-Seal™ during the study period from January 2016 - September 2024.  Procedural data, pre and post-procedural hemoglobin and hematocrit levels were analyzed.  Technical success was defined as the successful deployment of the device with immediate hemostasis and no evidence of bleeding on post-procedural imaging.  The authors report 100% technical success.    However, Please note the following points:

  1. Was any statistical analysis to determine the power of the study.  How did the authors determined the number 20?
  2. Given that this was a retrospective study, why more number of patients selected for the study.
  3. In the Abstract, end of last line, just p is mentioned without any value.  
  4. Page 4, para 2:  Please mention the dosage of warfarin given and then mention, the INR and PT values.  
  5. "In this technique the procedure was carried out while the patient was under IV sedation."  Please mention the drug and dosage.
  6. Given that Angio-Seal™ can be associated with both thrombotic and bleeding complications and the authors have made sure that there was no bleeding associated, but for thrombotic complications have the authors tested for markers of coagulation activation such as thrombin generation, d-dimer levels and F1.2 levels?. No significant drop in hemoglobin or hematocrit values and other imaging results may ensure that there were no bleeding complications, however, for any coagulation activation, results of thrombin generation, d-dimer and F1.2 levels are necessary before reporting 100% technical success claim.  Please explain.

Author Response

Comment 1:

Was any statistical analysis to determine the power of the study? How did the authors determine the number 20?

Response 1:
We appreciate the reviewer’s insightful question regarding sample size justification. As this was a retrospective study, a formal power analysis was not performed prior to data collection. Instead, we included all eligible patients who underwent percutaneous transhepatic portal venous interventions with Angio-Seal™ tract closure at our institution between January 2016 and September 2024. The total number of cases was limited by the rarity of such interventions and the selective use of this closure technique in our clinical practice. Nevertheless, based on the observed complication-free technical success rate (100%) and the consistent procedural protocol, we believe this cohort provides meaningful preliminary data. A post hoc power calculation based on the significant drop in hemoglobin and hematocrit values could be considered, but our main goal was to share initial feasibility and safety results. We acknowledge the need for larger, prospective studies to validate these findings and establish statistical robustness.

Revision Note:
No changes were made in the manuscript text regarding this comment.

Comment 2:

Given that this was a retrospective study, why were more patients not selected for the study?

Response 2:
We appreciate the reviewer’s comment regarding the sample size. Between January 2016 and September 2024, a total of 48 percutaneous transhepatic portal vein access procedures were performed at our center. However, only 20 patients were included in the present study due to specific constraints. During certain periods within this timeframe, our institution experienced difficulties in obtaining the Angio-Seal™ device due to supply chain and distributor-related issues. Consequently, tract closure in some cases was performed using alternative methods such as coils, glue (N-butyl cyanoacrylate), or a combination of these embolic materials. Due to these external limitations in device availability, we were unable to standardize the closure technique across all patients and thus could not expand the cohort beyond the included 20 cases. This limitation was beyond our control and reflects the real-world challenges associated with device-dependent procedural standardization in interventional practice.

Revision Note:
No changes were made in the manuscript text regarding this comment.

Comment 3:

In the Abstract, end of last line, just p is mentioned without any value.

Response 3:
Thank you for pointing this out. We have reviewed the original version of the manuscript and confirmed that the p-values were correctly stated in the abstract as: (p < 0.001) for the decline in hemoglobin and (p = 0.001) for the decline in hematocrit. It is likely that this issue resulted from a formatting or file conversion error during the upload or review process. We will make sure the correct version is reflected in the final proof.

Revision Note:
Correct p-values have been preserved and verified in the Abstract section.

Comment 4:

Page 4, para 2: Please mention the dosage of warfarin given and then mention, the INR and PT values.

Response 4:
None of the patients in this study were on warfarin therapy at the time of the procedure. The mean pre-procedural INR was 1.33 (range: 1.05–2.54), and the mean prothrombin time (PT) was 23.1 seconds (range: 12.2–90.2), as indicated in the manuscript. Although most patients had INR and platelet values within or near normal limits, in cases with elevated INR or thrombocytopenia, coagulation parameters were corrected with supportive treatment such as fresh frozen plasma, vitamin K, or platelet transfusion prior to the procedure.

Revision Note:
This information has been added to the Materials and Methods section, under the paragraph describing INR, PT, and pre-procedural laboratory evaluation.

Comment 5:

"In this technique the procedure was carried out while the patient was under IV sedation." Please mention the drug and dosage.

Response 5:
Thank you for your insightful comment. All procedures were performed under intravenous sedation administered by an anesthesiologist. The sedation regimen, selected and titrated at the anesthesiologist's discretion, typically included midazolam (0.02–0.05 mg/kg) and fentanyl (1–2 mcg/kg). In some cases, propofol (0.5–1 mg/kg bolus) was used as needed for deeper sedation.

Revision Note:
This information has been incorporated into the Materials and Methods section under the procedural technique description.

Comment 6:

No significant drop in hemoglobin or hematocrit values and other imaging results may ensure that there were no bleeding complications; however, for any coagulation activation, results of thrombin generation, d-dimer and F1.2 levels are necessary before reporting 100 % technical success claim. Please explain.

Response 6:
Thank you for drawing attention to this point. In the original manuscript we used “technical success = successful device deployment with immediate hemostasis and no imaging-proven bleeding.” We recognise that (i) this definition does not capture sub-clinical coagulation activation and (ii) absence of bleeding might, in some patients, have been achieved even without tract closure. Consequently, “100 % technical success” needs to be interpreted strictly within the confines of that procedural definition.

Because our study is retrospective, includes a limited sample, and lacks a control group in which the tract was left untreated or closed by another method, we cannot assert that Angio-Seal™ alone was responsible for the absence of bleeding in every case. The statement has therefore been qualified as follows:

  • Results section (first sentence): “Technical success, as defined in this study (see Methods), was achieved in all 20 procedures (20/20, 100 %).”
  •  
  • In addition, a paragraph has been added to the Limitations section to acknowledge that sub-clinical thrombotic activation was not assessed and that our success rate reflects the chosen, procedure-based definition rather than an absolute measure of clinical efficacy.

Revision Note:
The wording “100 % technical success” has been restricted to the specific procedural definition, and a caveat regarding the absence of systemic coagulation markers and a control group has been inserted into both the Results and Limitationssections.

Reviewer 2 Report

Comments and Suggestions for Authors

Nice review

  1. Can you please talk about length of deploying sheath of angioseal and size of the patient considerations.
  2. Can you please discuss the size of foot plate of angioseal and site of access into the portal vein consideration. Like peripheral vs cental portal veins in terms of vessel size.
  3. Line 35-36, you mention about 30% bleeding risk when tract not closed. Can you please discuss the bleeding risks when tract is closed with coils, gelform etc. 
  4. Can you please discuss cost value analysis of using angioseal vs other closure agents.
  5. Can you please add contraindication to the use of angioseal in patients with collagen allergy.
  6. 6. Can you please add a table detailing all the closure devices available in the market with pros and cons and their sizes and method of delivery.

Author Response

Comment 1:
Can you please talk about length of deploying sheath of Angio-Seal and size of the patient considerations?

Response 1:
We thank the reviewer for this important technical observation. The Angio-Seal™ VIP device is provided with a deployment sheath approximately 11 cm in length. In our adult cohort, this sheath length was sufficient to accommodate the entire skin-to-portal vein tract, which typically ranged between 6 and 10 cm, as confirmed by pre-procedural imaging. No patient in our series required the use of extension sheaths or alternative access approaches.

However, in pediatric patients or individuals with reduced liver volume, the hepatic parenchymal tract may be shorter than the standard 11 cm sheath length. In such cases, complete intrahepatic deployment of the collagen plug and anchor system could be compromised, potentially resulting in peritoneal deployment or bleeding complications. Therefore, careful pre-procedural assessment of cross-sectional imaging (CT or MRI) is crucial to confirm anatomical suitability for Angio-Seal™ deployment.

Revision Note:
This consideration has been added to the Discussion section to emphasize the importance of sheath length relative to patient anatomy, particularly in small-statured or pediatric patients, with supportive references [8,19].

Comment 2:

Can you please discuss the size of foot plate of Angio-Seal and site of access into the portal vein consideration. Like peripheral vs central portal veins in terms of vessel size.

Response 2:

We appreciate the reviewer’s insightful question. In our study, percutaneous access was performed through the right side in 80% of the patients, and in all cases, a peripheral portal vein branch was deliberately selected for puncture. This strategy was based on both anatomical suitability and procedural safety, as peripheral branches offer better parenchymal support for device deployment and reduce the risk of migration or malposition, which may be encountered in larger-caliber and straighter central portal vein segments.

The Angio-Seal™ VIP device, used in both 6F and 8F sheath-compatible versions in our series, features a T-shaped bioabsorbable anchor designed to lodge within the vessel lumen and secure tract closure in combination with a collagen plug. According to Hon et al. (2009), the anchor in the 6F system measures approximately 10 mm in length and 1.2 mm in width, dimensions that were deemed compatible with the size of peripheral portal vein branches accessed in our cases. Although manufacturer specifications for the 8F anchor are not publicly detailed, it is generally accepted that the anchor is proportionally larger to accommodate the wider sheath size.

The selection of sheath size—and thus the corresponding Angio-Seal™ device—was tailored to the procedural requirements, with larger sheaths (up to 8F) used in interventions involving stent placement or thromboaspiration. Regardless of sheath size, pre-procedural imaging was carefully reviewed to ensure that the accessed vessel had a sufficient diameter and that the hepatic parenchyma would provide adequate support for anchor deployment.

Revision Note:

This clarification has been incorporated into the Discussion section, elaborating on the rationale for peripheral access site selection and device compatibility based on anatomical considerations and anchor dimensions【Hon et al., 2009】

Comment 3:

Line 35–36, you mention about 30% bleeding risk when tract not closed. Can you please discuss the bleeding risks when tract is closed with coils, gelform etc.?

Response 3:

We thank the reviewer for this valuable comment. While the initial version of the manuscript focused on bleeding risks in the absence of tract closure, we fully agree that a comparative discussion of bleeding risks associated with commonly used embolic agents is warranted.

As requested, we have expanded the Discussion section to include relevant data from recent literature. In particular, we referenced the study by Marra et al. (2022), which retrospectively analyzed outcomes from 220 percutaneous portal vein catheterizations across two centers. That study compared embolization techniques including coils, gelatin sponge, and NBCA. Although all methods showed high technical success and no statistically significant differences in post-procedural bleeding rates, bleeding events were more frequently reported in the coil and non-embolization groups. Furthermore, one case of non-target embolization was observed following NBCA use, but without clinical consequences.

In contrast, our series—using a single Angio-Seal™ device without additional embolic material—demonstrated no clinical or radiologic evidence of bleeding or non-target embolization. These findings suggest that, in carefully selected patients with adequate hepatic parenchymal tract length, single-device closure may offer comparable or potentially improved hemostatic control relative to conventional embolization techniques.

Revision Note:

A comparative paragraph discussing the bleeding risks associated with coils, gelatin sponge, and NBCA was added to the end of the Discussion section, including citation of the study by Marra et al. (2022) to address the reviewer’s concern.

Comment 4:
Can you please discuss cost value analysis of using Angio-Seal vs other closure agents?

Response 4:
We thank the reviewer for this important comment. While a formal cost analysis was not part of our study design, relevant comparisons can be drawn based on procedural complexity and material usage reported in the literature. Zhang et al.  compared NBCA and coil-based embolization, reporting a mean NBCA volume of 0.35 ± 0.07 mL and an average of 1.2 ± 0.4 coils per case. These techniques often require multiple components, extended fluoroscopy time, and high operator precision, potentially increasing the total procedure cost.

In contrast, our approach utilized a single Angio-Seal™ device per case without adjunctive embolization. This standardized method enabled rapid hemostasis and eliminated the need for additional materials or extended manipulation. Although the device may be more expensive per unit, its use may reduce overall costs by shortening procedure time and minimizing complications or follow-up interventions.

Revision Note:
A concise paragraph discussing the cost-efficiency of Angio-Seal™ versus coils and NBCA has been added to the Discussion section, referencing Zhang et al.

Comment 5:
Can you please add contraindication to the use of Angio-Seal in patients with collagen allergy?

Response 5:
We thank the reviewer for this important suggestion. As recommended, we have now clarified that the Angio-Seal™ device contains a bioabsorbable collagen sponge derived from bovine sources and should not be used in patients with known hypersensitivity to collagen. A corresponding statement has been added to the Materials and Methods section:

“It is important to note that the Angio-Seal™ VIP device contains a bioabsorbable collagen sponge derived from bovine sources (Achilles tendon). Therefore, its use is contraindicated in patients with known hypersensitivity to bovine-derived materials or collagen. All patients included in this study were screened for any history of collagen allergy prior to device deployment.”

We believe this addition improves the safety clarity of the manuscript and thank the reviewer for helping to strengthen this aspect.

Comment 6:
Can you please add a table detailing all the closure devices available in the market with pros and cons and their sizes and method of delivery?

Response 6:
We sincerely appreciate the reviewer’s thoughtful suggestion. We agree that a comprehensive table comparing all currently available vascular closure devices (VCDs)—including their mechanisms, sizes, delivery methods, and clinical advantages or limitations—would be informative to the broader readership.

However, given the specific and focused nature of our study, which primarily evaluates the technical feasibility and safety of the Angio-Seal™ device in percutaneous portal venous tract closure, a detailed comparative table encompassing the full spectrum of VCDs would extend beyond the methodological intent and scope of this work.

Nonetheless, we have addressed relevant alternative closure devices and their mechanisms within the Discussion section and cited recent literature where broader overviews of VCD technologies are provided. We hope this targeted approach maintains the clarity and focus of the manuscript while still offering readers a meaningful context.

Should a broader comparative review be warranted, we believe it would be better suited for a dedicated article focusing specifically on closure device technologies across various anatomical applications.